# Occupational Burnout in Healthcare Workers, Stress and Other Symptoms of Work Overload during the COVID-19 Pandemic in Poland

**DOI:** 10.3390/ijerph20032428

**Published:** 2023-01-30

**Authors:** Zbigniew Izdebski, Alicja Kozakiewicz, Maciej Białorudzki, Joanna Dec-Pietrowska, Joanna Mazur

**Affiliations:** 1Department of Biomedical Aspects of Development and Sexology, Faculty of Education, Warsaw University, 00-561 Warsaw, Poland; 2Department of Humanization of Health Care and Sexology, Collegium Medicum, University of Zielona Góra, 65-046 Zielona Góra, Poland

**Keywords:** burnout, stress, overwork, COVID-19, mobbing, health care workers

## Abstract

This study explored the level and selected determinants of burnout among five groups of healthcare workers (physicians, nurses, paramedics, other medical and nonmedical staff) working during the COVID-19 pandemic in Poland. This cross-sectional study was conducted from February to April 2022, with the use of a self-administered mostly online survey. The BAT-12 scale was used to measure burnout, and the PSS-4 scale was used to measure stress. The sample was limited to 2196 individuals who worked with patients during the COVID-19 pandemic. A series of multivariate logistic regression models with three to nine predictors was estimated. The prevalence of burnout ranged from 27.7% in other nonmedical staff to 36.5% in nurses. Adjusting for age and gender, both physicians (*p* = 0.011) and nurses (*p* < 0.001) were at higher risk of burnout. In the final model, elevated stress most likely increased the risk of burnout (OR = 3.88; 95%CI <3.13–3.81>; *p* < 0,001). Other significant predictors of burnout included traumatic work-related experience (OR =1.91, *p* < 0.001), mobbing (OR = 1.83, *p* < 0.001) and higher workload than before the pandemic (OR = 1.41, *p* = 0.002). Only 7% of the respondents decided to use various forms of psychological support during the pandemic. The presented research can contribute to the effective planning and implementation of measures in the face of crisis when the workload continues to increase.

## 1. Introduction

Work is an important aspect of people’s lives. It impacts their everyday functioning and may be a source of great happiness and success. However, it can also be a source of stress and anxiety, leading to emotional problems and depression [1]. Over the past decades, work-related burden has been on the increase. It can reduce workers’ self-efficacy and impact their health, thereby causing professional burnout [2,3]. The term ‘burnout’ was first explored in 1974 by Herbert Freudenberg, who described it as the emotional and psychological stress experienced by workers [4]. Since then, the concept of burnout has been conceptualized as workplace stress in every occupational context [5].

The term burnout can be used as a shortcut for a psychological syndrome encompassing three dimensions: emotional exhaustion (EE), depersonalization (DP) and decreased sense of personal accomplishment (PA), according to the Revision of the International Classification of Diseases (ICD-11) [6]. In this three-dimensional model, EE refers to feelings of work overload and depletion of one’s emotional resources; DP refers to one’s negative response to other people, both colleagues and patients, in a cynical way; PA is the tendency to negatively evaluate the worth of one’s work and feel insufficient in regard to the ability to perform one’s job. Therefore, healthcare providers experiencing DP can become insensitive and less empathetic when managing their patients, creating distance in their provider–patient contact. DP may negatively impact professionalism. Deficits in PA may lead to feelings of incompetence in professional efficacy, which may impair healthcare professionals’ ability to accomplish their tasks. Burnout often involves feelings of a lack of control and a diminished sense of PA at work, which further reinforces a sense of being underestimated [7,8]. According to the research, burned out individuals are more withdrawn from the job and less likely to take up new challenges coming their way at work; psychological factors, e.g., being less extroverted, are seen as major burnout triggers [9,10,11]. Individuals experiencing burnout tend to do the bare minimum to complete a task. They do only what is necessary, which in turn adversely impacts their professional development, self-efficacy, and the quality of service they provide [12,13]. A systematic literature review suggests that burnout affects healthcare services and leads to an increased sense of work-related stress, greater pressure, excessive workload and organizational chaos [14].

Burnout may affect professional groups in different sectors. It is estimated that 13%-27% of the labor force has been affected [15]. Healthcare workers are particularly susceptible to burnout [16], especially given the fact that clinical practice is an important burnout trigger due to the ongoing contact with patients and suffering [17]. Burnout among healthcare workers adversely affects not only the department they work in but also their performance and the functioning of the entire healthcare system. It increases the risk of medical errors and adversely affects patients’ safety [18]. Higher levels of burnout are also associated with greater patient dissatisfaction and increased patient and family complaints [19]. An important implication for the healthcare system is also the fact that burned out physicians take early retirement more often, which, in turn, may delay or even prevent patient access to the most experienced physicians and stretch the waiting time for treatment. Moreover, incidence of depression, drug and alcohol abuse, and suicide is significantly higher in physicians experiencing burnout [20].

Burnout in healthcare professionals is associated with work-to-family conflict, unrealistic expectations of patients, an ongoing pressure on continuous learning, long working hours, excessive bureaucracy, organizational issues, poor communication among healthcare professionals, and personal issues [20]. An increased amount of time spent on the job also poses a greater risk for burnout. In a study of 7905 surgeons, 30% of surgeons working less than 60 h a week met the criteria for burnout, while among those working over 80 h the burnout rate was 50% [18].

A hospital, as a healthcare institution, should provide its workers with adequate professional and personal support [21]. An integrative review showed the importance of enhanced social support and the important role hospital authorities play in promoting support for burned out employees [22]. Other studies found that respect of and concern for others, as well as recognition for employees, were among those attributes that showed higher rates of work-related satisfaction in employees, which may be relevant for preventing burnout [23].

Work-related stress among healthcare workers has become a significant health issue not only for employees but also for the entire economy [24]. The COVID-19 pandemic negatively affected the mental well-being of employees in various work settings and professional groups [25,26]. Anxiety, anger, frustration, post-traumatic stress symptoms and greater stress were cited as the effects of the pandemic, while at the beginning of the pandemic in Poland the average Cohen’s stress index was increased −6.38 ± 2.94 [27,28,29,30]. Undoubtedly, healthcare workers played a key role in fighting the consequences of the pandemic but, at the same time, their professional and private lives were strongly disrupted. Their daily workload increased significantly, and many were redeployed to work in clinical areas outside their usual practice, which also entailed frequent changes in their roles and duties [31,32]. Many had to give up their social life to protect their families, themselves and their patients from coronavirus.

Considering the knowledge available to date, there is still a lack of studies that would address occupational burnout with various aspects associated with work, stress and a profession during the COVID-19 pandemic in Poland. Extensive research that was conducted in 114 facilities in Poland allows us to evaluate the issues of burnout across the country and professional groups. This is one of the very few large-scale studies on occupational burnout among healthcare workers in the context of the COVID-19 pandemic. Given the perspective of burnout prevention in healthcare workers, and when trying to gain insights into the problem, it is important to pay attention to the mental health of health workers, including stress, profession, workplace mobbing and workplace trauma as well as financial status and years of service. It can be hypothesized that the incidence of burnout varies across healthcare professions and is a factor of work-related aspects and the profession itself. It can be expected that individual professional groups and the level of stress significantly impacts burnout.

This study analyzed the level of burnout among healthcare workers in medical units during the COVID-19 pandemic in Poland. The study assessed the extent to which certain sociodemographic characteristics, perceived stress and other work-related demands increase the risk for burnout. Moreover, the study showed how often professional psychological assistance and less formal support groups were used by healthcare workers, relative to the level of burnout.

## 2. Materials and Methods

### 2.1. Study Design

The survey was conducted as part of a larger project on the humanization of medicine and clinical communication from 2 March to 28 April 2022, and a cross-sectional design was used. One hundred and fourteen health care units, including 94 hospitals, consented to participate in the project. This study used a self-administered online survey (CAWI) technique registered with a research panel developed by Research Collective sp. z o.o., a company based in Poland. Some data were collected using the pen-and-paper (PAPI) technique. As a result, 2340 questionnaires were obtained from medical personnel, including 249 paper-based questionnaires.

The questionnaire contained questions regarding healthcare workers’ evaluations of various aspects of their relationships with coworkers, patients and their families, as well as questions about certain aspects of their own lives, and the impact of the COVID-19 pandemic on their evaluations. The respondents provided answers to close-ended questions, mainly on nominal or ordinal scales and visual analogue scales. The average time to complete the online questionnaire was 23.74 min, with the median reaching the value of 20.75 min. The questionnaire could have been left with incomplete answers at any time, without giving any reason and without any consequences. Respondents completed the questionnaire voluntarily, with full information on what the survey was about and what type of study it was. At the same time, consent to participate in the survey was given by choosing ‘yes’ or ‘no’ answers on the computer screen, due to the online nature of the survey.

### 2.2. Assessment of Burnout

The Burnout Assessment Tool (BAT-12) was used as a dependent variable. The Polish version of the BAT-12 was developed for the purposes of this project, after having obtained the consent from W.B. Schaufeli, the author of the instrument. The BAT has a long version that consists of 23 items. Translation and back translation were performed for the short form BAT-12. The BAT was developed to measure burnout as a general score, and to assess each of its four core dimensions (exhaustion, mental distance, cognitive impairment and emotional impairment) and its three secondary dimensions (psychological distress, psychosomatic complaints and depressed mood). This study focused only on the above core BAT-12 dimensions, with 3 items in each one [33]. Participants answered the items on a scale of 1 (never) to 5 (always). The original BAT study concluded that the internal consistency of BAT-12 was very good (α > 0.92) but somewhat lower, by definition, than the internal consistency of BAT-23 (α > 0.97) [34].

The mean scores on the BAT scales are calculated by adding the scores on all items of a particular subscale and then by dividing this sum by the number of items. According to norms provided for Belgian and Dutch populations, the cut-off points of 2.54 and 3.02 could be applied to define the risk for burnout and when burnout is most likely, respectively [33].

In the study sample (N = 2196), Cronbach’s alpha was 0.926 for the BAT-12. The values of model fit indices (CFA) are as follows: RMSEA = 0.0945 (90%CI 0.0895–0.0997), SRMR = 0.0491, CFI = 0.951, TLI = 0.933.

### 2.3. Assessment of Stress

Another standardized scale measuring the perception of stress, the short 4-item Perceived Stress Scale (PSS-4), was used as a key independent variable. The Perceived Stress Scale (PSS), also known as Cohen’s scale [35], was used to measure stress levels. Three versions of the PSS (PSS-14, PSS-10, and PSS-4) are available, comprising 14, 10, and 4 items, respectively. While the PSS-10 is highly recommended, some authors have objected to the 4-item version [36]. However, it works well in multi-threaded questionnaires since it allows us to reduce the time needed to collect data. The PSS-4 comprises four questions asked from the perspective of the past month experience. An example of such a question is: ‘In the past month, how often have you felt difficulties were piling up so high that you could not overcome them?’ Five categories of answers were provided, i.e., from never to very often. The answers were coded from 0 to 4 for negative statements and from 4 to 0 for positive responses. The summary index takes the range from 0 to 16 points, where a high score means a significant stress intensity. No clinically validated cut-off point exists that would identify high stress. Only average values are available in the literature for PSS-4. Sometimes, a cut-off value of 6 or above is suggested, according to the norms in British studies [37]. For the purposes of this study, it was assumed that score values of 6 ≥ (greater than or equal to 6) indicate increased or high stress coded into a binary variable as ‘1′.

### 2.4. Other Variables

An important set of questions referred to professional support used by healthcare workers in mental health crisis. Psychological, psychiatric and support group assistance (formal and informal) were analyzed independently, allowing the respondents to refuse to answer the question. Similar questions were asked about the existing need to seek these types of support. Derivative variables defining the use or willingness to use any proposed types of assistance were created.

The questions concerning the financial situation of the respondents asked which of the following statements best describe the financial situation of their household, with three response categories ranging from ‘we are unable to cover essential expenses with income’ to ‘we are able to put aside/invest part of our income’. For the purposes of this analysis, this question was categorized into three levels of discrimination–situation rather poor, fair and good. This question has been used many times in the national household and our own [38,39] surveys.

### 2.5. Statistical Analyses

At the preliminary analysis phase, the psychometric properties of the BAT-12 scale were examined. The Cronbach’s alpha coefficient was used to estimate the internal consistency of the data in the BAT-12 scale. Generally, values for Cronbach’s alpha above 0.70 are considered to indicate a reliable set of items [40]. The normal distribution of the BAT-12 was verified with the Kolmogorov–Smirnov test (K-S test). Descriptive analysis was performed using the mean and standard deviation. The distribution of mean scores for the four dimensions of BAT-12 was defined as burnout.

The responses to the individual questions were compared for different professions. The responses were examined separately for these professions using the chi-square test (categorized variables). The results of the chi-square test (chi-sq.) are presented along with the degrees of freedom (d.f.) and the significance level (p).

To allow the comparison of the distributions of BAT-12 scores, a non-parametric Kruskal–Wallis test was used, taking into account failure to meet the assumption of normal distribution. A non-parametric post hoc test was used to compare the groups pair by pair. Within the multivariant regression analyses, binary logistic regression outcomes were shown, using the entire categorized BAT-12 index as the dependent variable. Lack of burnout was coded as ‘0′, while the risk for and highly likely burnout were coded into ‘1′. A total of eight explanatory variables were included, with response categories given further in the table characterizing the sample. The dichotomous variables (0–1) were gender, workplace mobbing, traumatic experiences and levels of stress. The ordinal variables included profession, years in the profession, work overload during the COVID-19 pandemic, and the financial and family situation. The only continuous variable was age in years. In the first regression model, we measured the impact of six factors, controlling for gender and profession. In the next step, all nine independent variables were added to the regression model. Logistic regression outcomes have been presented as odds ratios (OR) with 95% confidence intervals (95% CI). At the stage of simple comparisons of work groups, a model with three dependent variables (work group, gender, age) was also estimated.

The Statistical Package for the Social Sciences (SPSS) 27.0 (IBM Corp. Released 2020. IBM SPSS Statistics for Windows, Version 27.0. IBM Corp., Armonk, NY, USA), Amos 26.0 (IBM, Armonk, NY, USA) and Jamovi version 2.3. (https://www.jamovi.org) were used for data analysis. The significance level was set at *p* < 0.05.

## 3. Results

### 3.1. Sample Characteristics

For this study, the sample was limited to 2196 healthcare workers who worked with patients on a day-to-date basis during the COVID-19 pandemic. The sample characteristics have been presented in Table 1. Our study sample is not gender balanced, with more female (81.2%) than male participants (18.8%). Most employee respondents had worked 10 or more years (72.5%), 9.8% for 6-10 years, and 17.7% fewer than 6 years. The respondents were put into five groups, with the largest group comprising nurses (52.7%), followed by physicians (22.1%), other healthcare professionals (10.1%), non-medical professionals (7.9%), and paramedics (7.3%). The table also shows the frequency distribution of answers to the questions concerning workload, workplace mobbing and trauma, financial situation, and stress. Most respondents worked the same amount of time before and during the pandemic (58.9%), 34.7% worked more than before the pandemic, and 6.5% of the respondents worked fewer hours. The survey revealed that 7.2% of healthcare professionals experienced workplace mobbing very often or quite often, and 33% had a traumatic experience. The respondents described their material status answering the question *Which of the following statements best describes your household financial situation?*, given the following response categories: we are unable to cover essential expenses with income; we cannot afford many things but are able to cover essential expenses with income; we are able to cover essential expenses but are unable to bear higher expenses; we can bear higher expenses; we can cover all expenses and put aside/invest part of our income. The table presents summary answers and 28.8% of the respondents found their situation good, 51.5% fair, and 7.1% bad. Moreover, 55.1% of the study population had an increased/high level of stress. The sample characteristics lack data on workplace mobbing, workplace trauma, and financial situation due to the sensitive nature of the questions.

### 3.2. Occupational Burnout

The mean scores of occupational burnout measured with the BAT-12 varied across professional groups, ranging from 2.15 (0.69) to 2.30 (0.69), and were the highest in nurses. A total of 18.1% of nurses, 17.3% of physicians and 15.0% of paramedics were found to be at risk for burnout. The scores indicating a significant risk for occupational burnout were the highest in nurses (18.4%), followed by persons of non-medical professions (16.8%), physicians (15.8%), paramedics (14.4%), and persons undertaking other medical professions (13.2%). The groups differed significantly in terms of burnout levels (chi-sq = 17.719; d.f. = 8; *p* < 0.023), which is shown in Table 2. In logistic regression adjusted for gender and age only, taking other medical professions as the reference group, a significantly higher risk of burnout (risk and significant risk combined) was found in both physicians (OR = 1.625l; *p* = 0.011) and nurses (OR = 1.817; *p* < 0.001).

Table 3 presents the distribution of stress, mobbing and traumatic experience by profession. Increased and high levels of stress characterized, to a greater extent, all professional groups, with the highest rates for nurses (58.6%), followed by paramedics (56.9%), other medical professions (55.1%), non-medical professions (51.4%) and medical doctors (50.4%). The groups differed in terms of stress levels (chi-sq = 15.436; d.f. = 4; *p* < 0.004). Nurses (24.4%) and paramedics (20.3%) were most often exposed to mobbing. Workplace mobbing was also reported by 18.2% of those working in other medical professions, 17.9% of non-medical personnel, and least of all by medical doctors: 14.2%. The groups varied in terms of exposure to mobbing (chi-sq = 15.496; d.f. = 8; *p* = 0.05). The highest percentage of paramedics (41.3%) experienced workplace trauma, followed by nurses (36.2%). A total of 33.5% of physicians, 20% of other medical professionals and 19.1% of non-medical personnel also reported workplace trauma. The professions varied in terms of work-related trauma during the COVID-19 pandemic (chi-sq = 42.374; d.f. = 4; *p* < 0.001).

Table 4 shows mean indices for the individual dimensions of burnout by exhaustion, mental distance, emotional impairment and cognitive impairment. The Kruskal–Wallis test revealed differences across professional groups with respect to exhaustion (*p* < 0.001), mental distance (*p* = 0.003) and cognitive impairment (*p* = 0.022). Post hoc pair by pair comparisons for exhaustion showed statistically significant differences between professional groups: nurses–other medical professions (*p* = 0.004); paramedics–nurses (*p* = 0.013); non-medical professions–nurses (*p* = 0.012), physicians–nurses (*p* = 0.016). A post hoc pairwise analysis for mental distance revealed differences between professional groups: other medical professions–physicians (*p* = 0.009); other medical professions–nurses (*p* < 0.001), non-medical professions–nurses (*p* = 0.037). Additionally, for cognitive impairment, post hoc pair by pair comparisons revealed differences between professional groups: other medical professions–physicians (*p* = 0.022), other medical professions–nurses (*p* = 0.008), non-medical professions–nurses (*p* = 0.028). Figure 1 presents the distribution of individual burnout dimensions by five professional groups.

### 3.3. Determinants of Occupational Burnout

Table 5 presents prevalence estimates of nurses’ burnout (understood as a group of persons at risk for burnout and of high risk for burnout) depending on years of service, workload, workplace mobbing, workplace trauma, financial situation and stress. The results of a logistic regression analysis showing the impact of a selected characteristic adjusted only for gender and professional group are also presented. The highest rate of burned out persons was found in those with 6–10 years of service (41.4%). The odds ratio was 2.6, which means that those with 6–10 years of service were 2.6 times more likely to develop burnout. Moreover, 40.9% of the respondents who worked more during the pandemic than they did before the outbreak of COVID-19 were at risk of burning out and were 1.7 times more likely to develop burnout. Workplace mobbing, which was reported by 55.3% of the respondents, increased the risk for burnout by 3.5 times. Those who experienced trauma at work, which was reported by 45.9% of the respondents, were 2.6 more likely to develop burnout. Poor financial situation (51% of burned out subjects) was also a risk factor for burnout, increasing the risk by 3.4 times. The highest odds ratio was reported for increased and high stress, increasing the risk by 4.8 times.

The results of multivariate logistic regression, considering all predictors, including professional group, for the dependent variable occupational burnout (0—does not exist, 1—exists) identified predictors significant for burnout, which are presented in Table 6. Age (a continuous variable) is a significant predictor of the occurrence of burnout. Both working 6 to 10 years and working more than 10 years were significant predictors of burnout, in both cases accounting for about a twofold greater chance of burnout occurrence. The chance of experiencing burnout increases for those who are physicians, with the odds ratio for the reference group (other medical profession) increasing by 1.8 times, while for the nursing group the result is on the border of statistical significance (*p* = 0.068) and increases by 1.4 times for this occupational group. Those working more during the pandemic period had a burnout risk 1.4 times higher than those working the same amount. Additionally, it can be noted that those who experienced mobbing were almost 2.5 times more likely to suffer from burnout, and those who had a traumatic experience were 1.9 times more likely. With regard to the traumatic event, those who refused to answer or did not know also had a risk of burnout higher by almost 1.5 times. For those declaring a bad and average financial situation, it was more than two times higher for a bad situation and almost 1.5 times higher for an average situation. A nearly four times higher probability of burnout is reported for those experiencing elevated and high stress. The results of the study also indicate that gender did not have a statistically significant effect on the likelihood of burnout.

### 3.4. Using Professional Help (Psychological, Psychiatric and Support Group)

Because of the pandemic, 86 persons (3.9%) started using psychological help, 53 (2.4%) psychiatric, and 38 (1.7%) a support group assistance. At present, 254 respondents (11.6%) would like to use psychological help, 48 (2.5%) psychiatric, and 77 (3.5%) would opt for a support group. Given the entire study sample (N = 2196), Figure 2 presents the percentage distribution of persons willing to use and already receiving help by profession.

Concerning the levels of burnout, the largest group using professional help included persons at high risk for burnout—22.3%. Exactly 10% of respondents at risk for burnout also used professional assistance, with 4% of respondents with no symptoms of burnout also using such help. With regard to those who would be willing to seek such assistance, distributions also differed depending on the level of burnout. Almost half of the respondents at the highest risk for burnout (49.9%) would like to seek professional help. A total of 20.1% of those at risk for burnout would also be willing to seek help, with 8.3% of those who were not burned out wanting to do the same (Table 7).

## 4. Discussion

This research is unique in terms of its scale and sample. It was conducted in 114 healthcare units in all voivodships (NUTS2) in Poland. The research findings include 2196 various medical and non-medical professionals (physicians, nurses, paramedics, other medical and non-medical professionals) who worked with patients on a day-to-day basis during the COVID-19 pandemic in Poland.

Numerous studies carried out before the pandemic show that healthcare workers are among professions definitely at the highest risk for burnout (as are teachers and lawyers). Health professions are subject to greater burnout, particularly due to the constant exposure to mental and physical suffering or death in the workplace [41]. Apart from the individual consequences of burnout on healthcare workers, equally adverse effects of burnout include those affecting the quality of work of a given medical unit and the entire healthcare system. These include a decreased workforce efficiency, greater job turnover and leaving the profession, a greater number of medical errors, limited or difficult access to healthcare services, reduced confidence in healthcare, poor quality of healthcare services and low patient satisfaction with those services, toxic work environment, rising healthcare costs, higher rates of early retirement or sick leave on health grounds, increased healthcare costs, and financial losses due to these consequences for the entire healthcare system [42].

The pandemic, additionally, exacerbated the situation and considerably forced the healthcare system to take appropriate (oftentimes new and additional) measures to fight the pandemic, limit its negative consequences and manage excessive workloads of healthcare workers. At an individual level, the pandemic had a negative impact, increasing the number of hours worked by medical personnel. Moreover, isolation, being away from the family, fear of becoming infected or infecting others, losing close ones and coworkers, and implementing new and complex medical procedures also had a negative impact [41]. This resulted in increased negative mental and physical symptoms and, ultimately, higher risk for burnout in the health professionals concerned. Working in new and insecure conditions, an increased risk of infection, having to use protective equipment, and not knowing how to cope with the new pandemic definitely led to increased negative mental effects, which may have manifested themselves in increased anxiety, prolonged stress and even depression in the professional group under study. A global review of the psychological effects of COVID-19 conducted in 35 countries showed a high incidence of anxiety (from 22% to 33%) and depression (from 18% to 36%) among healthcare workers [43]. Additionally, ample works in the literature suggest that long-term and chronic stress have a negative influence on both psychological and physical health, leading to serious health conditions such as post-traumatic stress syndrome (PTSD) and burnout [44,45,46]. The existing data show that during the pandemic 70% of medical personnel suffered from anxiety and 50% of healthcare workers experienced depression [47]. Moreover, there were high rates of anxiety disorders and symptoms of depression in healthcare personnel during the pandemic, such as anxiety and depression disorders, especially in frontline healthcare workers (23% and 27%, respectively) [48]. Similar findings demonstrate that more than a third of the study clinicians experienced significant or moderate levels of constant stress or suffered from depression [49,50,51,52]. In other systematic review, the pooled prevalence of depression for frontline professionals was 43%, for nurses 25%, and 24% for medical doctors [53].

Our studies yielded similar results, where one in three respondents experienced a heavier workload during the pandemic than before the outbreak. Increased stress levels, role overload and lack of support may clearly contribute to an increased risk for burnout. Therefore, more than half of all respondents experienced a higher level of workplace stress than before the pandemic. In our sample, nurses were particularly hard hit. In the highly stressful and demanding settings of intensive care units, nurses suggest they experience a great deal of stress, burnout and low job satisfaction [54]. Staffing problems and work overload are among the most prevailing stressors among nurses [55]. According to many other scientific papers, the highest levels of burnout and burnout-related symptoms were found in nurses and frontline professionals [53,56]. Additionally, gender is one of the main predictors of early burnout and it mainly affects female healthcare professionals, who are more likely to develop work-related stress [48]. Other sociodemographic characteristics associated with high risk for burnout found in the literature include having less professional experience, not having children, being single, high levels of anxiety, symptoms of depression and a great deal of stress in females [41,57].

Regarding specific medical professions, most often high rates of burnout are more common among physicians and nurses. Apart from some individual issues, burnout is largely associated with external factors such as increased workload and ineffective interpersonal relationships [58]. According to the literature and findings, there is a large variability (0–80.5%) in the prevalence of burnout among physicians [59]. A study of 7288 physicians [60] revealed the highest rates of burnout in emergency room physicians (52%) and critical care physicians (50%), with the lowest being in psychiatrists (33%) and pathologists (32%). A meta-analysis of studies carried out before the pandemic found that the incidence of burnout associated with at least one of the three dimensions of burnout was reported in 30% of the respondents [61]. Additionally, studies in the UK show that, before the coronavirus outbreak, 50% of the surveyed medical professionals were emotionally exhausted [62,63] which, in addition to fatigue and frustration, was clearly indicative of burnout symptoms. Given the pandemic and all its consequences, the medical professions run the highest risk for burnout, especially over the long term [64].

In line with the structure of the BAT-12, the dimensions of burnout that were analyzed included exhaustion, mental distance, emotional impairment and cognitive impairment. Nurses obtained the highest mean sub-indices scores in three dimensions of burnout, with only emotional impairment being higher among physicians. In this one dimension of burnout, differences between professional groups were not statistically significant. Burnout symptoms were, relatively, less often experienced during the COVID-19 pandemic by members of other medical and non-medical professions. These groups were also less likely to report traumatic experiences during the COVID-19 pandemic.

For the sake of comparison, a multinational cross-sectional study of 3537 healthcare workers conducted early on in the COVID-19 pandemic found (using the OLBI) that 67% of the respondents showed symptoms of occupational burnout [31]. Our analyses thus show the need for a more cautious approach in drawing conclusions, as the studies on burnout were carried out in the world in different COVID-19 periods with different tools and the analyses also varied due to context factors.

Moreover, our findings show that severe stress and high levels of stress may almost quadruple the risk for burnout. Additionally, our research revealed a strong correlation between burnout and other determinants, such as greater than before workload, workplace mobbing and traumatic experiences, and also a poor financial situation. Excessive working hours and prolonged shifts may lead to physical and mental overload, which surely has an adverse impact on healthcare services and significantly contributes to burning out faster [14].

Apart from health and psychological problems, burnout may also lead to substance abuse, risky behaviors, increase in self-destruction, violence, or suicide incidence, and also absence from work or an increased number of medical errors [65,66,67,68,69].

To avoid the above effects, healthcare workers had not only to face the challenges of the new pandemic but also to adapt and act preventively. In the case of burnout experience, the research literature highlights the key issue of how to manage this problem: “*fixing the person versus fixing the job*?” [70]. Given the “fixing the person” approach, healthy behaviors, such as following a healthy diet, engaging in regular exercise, healthy sleep habits, being able to relax or avoiding addictive substances were particularly helpful in fighting the risks of burnout and its effects [71]. Preventive healthcare programs and measures, developing interpersonal skills, coping with stress, assertiveness training and problem-solving skills, time and work management training, mindfulness, self-understanding, relaxation and breathing techniques were also largely effective [72,73,74,75,76,77,78,79]. Balasubramanian et al. [71] also recommend a range of useful mobile applications and websites for self-help and self-care, which are also dedicated to healthcare workers. Of importance, too, are professional support and assistance. Our research findings show that 8% of the respondents decided to use professional support and assistance to deal with the adverse effects of working during the COVID-19 pandemic. Separate questions were asked about seeking psychological, psychiatric and support group assistance. For each type of assistance, the percentages of subjects was so low that we limited ourselves to analyzing them cumulatively. Using different methods of help was most common among nurses and paramedics, and relatively less common among the other three professional groups. Currently, such a decision would be considered by 18% of respondents, with paramedic professionals and other medical professionals being the most common.

On the other hand, coping with burnout syndrome may refer to the working environment and therefore include solutions in terms of “*fixing the job*”. Given this approach, any institutional arrangements and workplace programs can be helpful. In units where burnout is a potential problem, interventions to prevent or reduce it should be properly planned and designed [70]. They may include psychological assistance in the workplace, workshops and training courses for employees of a given unit, or mediation with a spokesperson or a person responsible for the employees. Additionally, Maslah highlights certain aspects that provide a framework for a healthy workplace: ‘The six positive elements that promote engagement and well-being can be defined as (a) a sustainable workload; (b) choice and control; (c) recognition and reward; (d) a supportive work community; (e) fairness, respect, and social justice; and (f) clear values and meaningful work’ [70]. Individual and systemic experiences with the COVID-19 pandemic have a chance to diagnose, verify, assess and implement these elements.

This study is one of the few that carried out comparative analyses of various professions, which allowed the investigation of individual professions relative to burnout and burnout levels against other healthcare workers. An additional advantage is that the study was nationwide in geographical (typological) terms and it accounted for the status of a healthcare institution (hospital and specialized outpatient clinics).

Attention should also be drawn to the use of various research tools and especially developed and adapted scales to measure the levels and risk of burnout among healthcare workers during the COVID-19 pandemic.

### Study Limitations

Particular attention shall be paid to some of the limitations of this research. It may be difficult to refer to and compare our findings with existing data in the literature for other countries. The differences arising from the pandemic situation in various countries, its severity and its negative effects and consequences do not allow for clear conclusions. Comparability is also affected by the tools used, variables taken into consideration in multivariate analyses and the research duration.

It should be noted that the research was conducted during the COVID-19 pandemic, which certainly presents a unique research sample; however, workload may have impacted the cognitive abilities of medical personnel at the time of completing the survey. Online surveys are prone to respondent bias. Particularly, those healthcare workers who agreed to answer the questions regarding their mental health may have been far more motivated to participate in the survey when in a poor mental condition, which may have to do with the misinterpretation of the presence and intensity of psychological symptoms.

Moreover, the survey allowed the respondents to refuse to answer questions that may have been seen as particularly intrusive, i.e., those that asked about workplace mobbing, traumatic experiences in the workplace and financial situation. Reasons for refusing to answer those questions could be that the respondents either did not want to touch on the above topics or did not know how to classify them. However, refusals were included as response categories for the model and for traumatic experience with regard to the reference variable (11.0%), and the result turned out to be statistically significant. Moreover, it can be seen that nurses were overrepresented (52.7%) and that the study was not gender balanced, with a significantly larger percentage of female respondents (81.2%). Although this advantage may distort the results, it should be noted, however, that it is actually a true picture of gender overrepresentation in healthcare professions [80]. Of relevant importance is also the fact that, thus far, Polish norms for the BAT-12 have not yet been developed.

Because of the cross-sectional nature of the study and a lack of longitudinal observation, it is not possible to make causal inferences between variables and about the identified long-term psychological effects.

Long-term implications for the mental health of healthcare workers and the effects of personal and organizational factors are worth further study. In future studies, it also seems reasonable to reach a consensus on how to classify the different levels of job burnout to make more accurate comparisons of the studies presented. Additionally, it would be worthwhile to consider examining job burnout in the area of various subspecialties [81], where, for example, physicians have less stressful working conditions, especially in relation to the COVID-19 pandemic during which some hospital departments had significantly higher workloads. Importantly, it is also worth considering a psychiatric evaluation of respondents before the survey. Indeed, pre-existing psychiatric illnesses or vulnerabilities may interact with the development of burnout during the COVID-19 pandemic [82]. Moreover, it is worth noting other distractions, such as personality, resilience, and empathy [83,84,85], which may also influence the burnout presented.

## 5. Conclusions

Since health workers are, largely, at the highest risk for burnout, it is necessary to take measures, particularly those targeting the most vulnerable medical professions (physicians and nurses), in order to minimize the effects of perceived stress. During medical studies and work, it is necessary to teach strategies for coping with difficult situations and arrange systematic workshops on the issues of burnout and coping with stress. Regarding the workplace, it is highly recommended to prepare an offer of professional support and assistance, widely available to healthcare workers in each institution, in particular to those who experience increased and high levels of stress. It is also necessary to make the offer of help to those who are facing the strongest predictors of burnout. Persons responsible for planning remedial and preventive actions in the workplace should pay particular attention to the reported cases of bullying and mobbing, workload and experiences of workplace trauma.

An effective and appropriate healthcare system response to the risk of burnout occurrence and actions taken to prevent and improve the situation, professional support accessible to healthcare workers and costs to support such actions may also be relevant.

This research, aiming to diagnose the situation during the COVID-19 pandemic associated with the risk of burnout among healthcare workers, can significantly contribute to the effective planning and implementation of the measures in question in the face of crisis, when the workload continues to increase.

## Figures and Tables

**Figure 1 ijerph-20-02428-f001:**
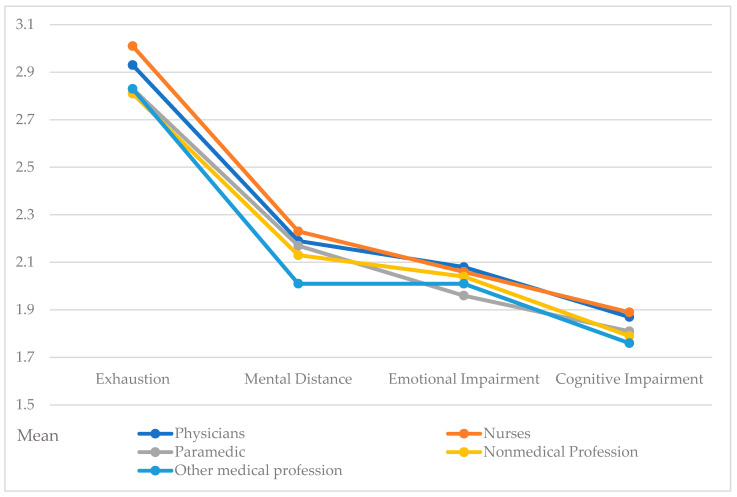
Occupational burnout profiles—mean percentage distributions for each dimension of occupational burnout by profession (N = 2196).

**Figure 2 ijerph-20-02428-f002:**
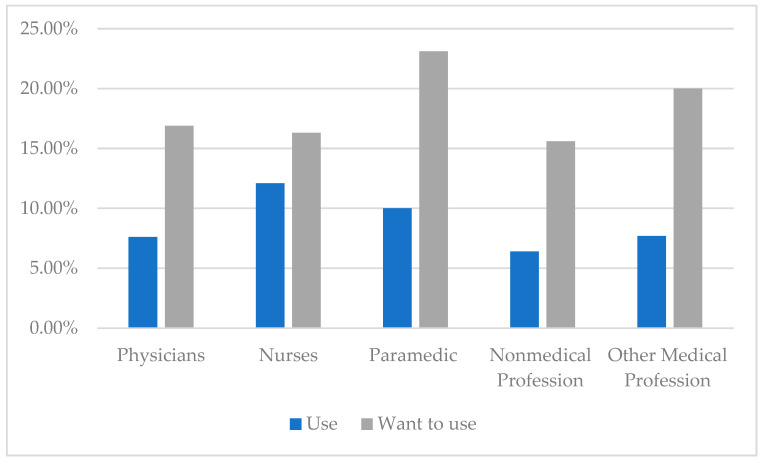
Percent distribution of respondents willing to use help or receiving help by profession (N = 2196).

**Table 1 ijerph-20-02428-t001:** Sample characteristics (all data presented as percentages; N = 2196).

Variable	Categories	Total N = 2196	Percentage
Gender	Male Female	413 1783	18.8% 81.2%
Duration of employment	Less than one year	56	2.6%
	1–2 years	137	6.2%
	3–5 years	196	8.9%
	6–10 years	215	9.8%
	More than 10 years	1592	72.0%
Profession	Physicians	486	22.1%
	Nurses	1157	52.7%
	Paramedics	160	7.3%
	Non-medical profession	173	7.9%
	Other medical profession	220	10.1%
Workload during the COVID-19 pandemic	The same amount of time as before (before the pandemic)	1293	58.9%
	Less than before the pandemic	142	6.5%
	More than before the pandemic	761	34.7%
Workplace mobbing experienced in the COVID-19 pandemic	Never	1379	62.8%
	Almost never	279	12.7%
	Sometimes	300	13.6%
	Quite often	77	3.5%
	Very often	37	1.7%
	Refused to answer	124	5.6%
Workplace trauma	Yes, I did.	725	33.0%
	No.	1091	49.7%
	I do not know.	241	11.0%
	Refused to answer	139	6.3%
Financial situation	Rather bad	155	7.1%
	Fair	1123	51.1%
	Good	632	28.8%
	Refused to answer	286	13.0 %
Stress level	Increased and high	1209	55.1%
	Low	987	44.9%

**Table 2 ijerph-20-02428-t002:** Mean occupational burnout scores by profession (N = 2196).

Profession	N	M (SD)	Levels of Occupational Burnout (%)			*p*
			Lack of Burnout	At Risk for Burnout	At a Significant Risk for Burnout	
Physicians	486	2.26 (0.66)	66.9	17.3	15.8	Chi-sq. = 17.719 d.f. = 8 *p* = 0.023
Nurses	1157	2.30 (0.69)	63.5	18.1	18.4	
Paramedic	160	2.19 (0.69)	70.6	15.0	14.4	
Non-medical profession	173	2.19 (0.74)	72.3	11.0	16.8	
Other medical profession	220	2.15 (0.69)	75.0	11.8	13.2	

**Table 3 ijerph-20-02428-t003:** The percentage of persons subject to workloads during the COVID-19 pandemic by profession (N = 2196).

	Increased and High Stress Level	Mobbing	Traumatic Event
	% (N)	% (N)	% (N)
Physicians	50.4 (245)	14.2 (69)	33.5 (163)
Nurses	58.6 (678)	20.3 (235)	36.2 (419)
Paramedics	56.9 (91)	24.4 (39)	41.3 (66)
Non-medical professions	51.4 (89)	17.9 (31)	19.1 (33)
Other medical profession	55.1 (106)	18.2 (40)	20.0 (44)
Chi-square test	15.436	15.496	42.374
DF ^1^	4	8	4
*p* ^2^	0.004	0.050	<0.001

^1^ DF—degrees of freedom; ^2^ significance level.

**Table 4 ijerph-20-02428-t004:** Mean occupational burnout profile scores by profession (N = 2196).

	Exhaustion	Mental Distance	Emotional Impairment	Cognitive Impairment
	M (SD)	M (SD)	M (SD)	M (SD)
Physicians	2.93 (0.85)	2.19 (0.85)	2.08 (0.77)	1.87 (0.75)
Nurses	3.01 (0.91)	2.23 (0.84)	2.06 (0.77)	1.89 (0.78)
Paramedics	2.83 (0.94)	2.17 (0.87)	1.96 (0.77)	1.81 (0.77)
Non-medical Professions	2.81 (1.01)	2.13 (0.93)	2.04 (0.76)	1.79 (0.83)
Other medical profession	2.83 (0.89)	2.01 (0.82)	2.01 (0.77)	1,76 (0.78)
DF ^1^	3	3	3	3
H-KW ^2^	17.855	16.175	4.580	11.474
*p* ^3^	<0.001	0.003	0.333	0.022

^1^ DF—degrees of freedom; ^2^ H-KW—Kruskal–Wallis, ^3^
*p*—significance level.

**Table 5 ijerph-20-02428-t005:** Prevalence of burnout by selected characteristics and the risk for burnout based on results of logistic regression analysis adjusted for gender and professional group (N = 2196).

			95% Confidence Interval	
	% Burnout	OR ^1^	Lower	Upper
Duration of employment				
Less than one year	21.4%	1		
1–2 years	30.7%	1.736	0.8296	3.632
3–5 years	27.6%	1.484	0.7262	3.033
6–10 years	41.4%	2.619 *	1.3033	5.263
More than 10 years	33.7%	1.681	0.8747	3.231
Workload				
The same as before the pandemic	29.3%	1		
Less than before the pandemic	30.3%	1.081	0.739	1.580
More than before the pandemic	40.9%	1.661 *	1.372	2.011
Mobbing				
No	27.7%	1		
Yes	55.3%	3.253 *	2.601	4.068
Refused to answer	35.5%	1.424	0.969	2.093
Traumatic event				
No	24.3%	1		
Yes	45.9%	2.599 *	2.120	3.185
Don’t know or refused to answer	35.5%	1.720 *	1.336	2.214
Financial situation				
Good	26.6%	1		
Poor	51.0%	3.379 *	2.314	4.933
Fair	37.0%	1.692 *	1.353	2.117
Refused to answer	24.8%	0.955	0.689	1.323
Stress level				
Low	16.2%	1		
Increased and high	47.6%	4.770 *	3.8845	5.858

^1^ Logistic regression adjusted for gender and professional group; * *p* < 0.05.

**Table 6 ijerph-20-02428-t006:** Final multivariate logistic regression model for the risk of burnout (N = 2196).

			95% Confidence Interval	
Predictor	*p*	Odds Ratio	Lower	Upper
Age	0.037	0.987	0.976	0.999
Gender				
Male—ref.		1		
Female	0.446	1.130	0.825	1.549
Duration of employment				
Less than one year—ref.		1		
1–2 years	0.089	1.584	0.932	2.692
3–5 years	0.316	1.286	0.787	2.101
6–10 years	0.004	2.095	1.262	3.447
More than 10 years	0.007	1.953	1.204	3.167
Profession				
Other medical profession—ref.		1		
Physicians	0.005	1.822	1.194	2.782
Nurses	0.068	1.420	0.974	2.071
Paramedics	0.655	0.885	0.518	1.513
Non-medical profession	0.548	1.116	0.706	1.925
Workload				
The same as before the pandemic		1		
Less than before the pandemic	0.737	1.074	0.708	1.630
More than before the pandemic	0.002	1.414	1.140	1.754
Mobbing				
No experience—ref.		1		
Experienced	<0.001	2.337	1.828	2.988
Refused to answer	0.563	1.134	0.741	1.735
Traumatic event				
No experience—ref.		1		
Yes	<0.001	1.910	1.522	2.397
Don’t know/Refused to answer	0.016	1.416	1.068	1.878
Financial situation				
Good—ref.		1		
Rather poor	<0.001	2.040	1.347	3.091
Fair	0.008	1.396	1.091	1.788
Refused to answer	0.239	0.826	0.577	1.184
Stress level				
Low—ref.		1		
Increased or high	<0.001	3.878	3.126	4.810

**Table 7 ijerph-20-02428-t007:** Percentage of people using professional help and those willing to use help now (psychological, psychiatric, support group) because of the COVID-19 pandemic by level of burnout (N = 2196).

	Lack of Burnout (N = 1463)		At Risk for Burnout (N = 362)		Burned Out (N = 371)	
	N	%	N	%	N	%
Receiving help	58	4.0%	36	10.0%	83	22.3%
Willing to use help	122	8.3%	73	20.1%	185	49.9%

## Data Availability

The data are owned by Warsaw University and are not to be made freely publicly available.

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
