# Peer review of "Occupational Burnout in Healthcare Workers, Stress and Other Symptoms of Work Overload during the COVID-19 Pandemic in Poland"

_ijerph, 2023, doi:10.3390/ijerph20032428_

Round 1

Author Response

Thank you for your very valuable comments. 
The responses are attached. 

Reviewer 2 Report

The manuscript's subject is of great importance and will attract a large readership. Therefore, the manuscript's findings, results, and "claims" must be far from speculation and thoroughly supported with appropriate references. Before I can give a more in-depth analysis and corresponding feedback on the Results section, the following points need to be addressed:

1)    The structure and the flow are very confusing in the Introduction; I suggest the authors rewrite this section while connecting the paragraphs and citing appropriate references.

2)    The discussion is very well-written and clear; thank you.

3)    A more concrete Conclusion section supported with clear statements is strongly advised.

4)    The main text needs to be significantly improved in terms of grammar, punctuation, and vocabulary. Therefore, a native speaker should edit the entire manuscript for grammar, punctuation, and clarity.

5)    Lines 56 – 58, please rephrase the sentence using appropriate spacing between words.

6)    The word "efficacy" is inappropriately used throughout the Introduction. The correct form of usage is self-efficacy; good use example within the manuscript's context would be: diminished/reduced self-efficacy.

7)    Line 61: the word "minimalistic" is not used in its proper context. Instead, authors could rephrase it as "individuals experiencing burnout tend to do the bare minimum to complete a task," and herein, the "silent quitting" concept could even be introduced.

8)    Lines 70 – 71, highly speculative claim, not supported with a sufficient number of references. These results simply "could" be due to the fact that the majority of healthcare professionals are female; thus, your claim does not have an objective basis. Please remove this sentence.

9)    Lines 512 – 519; statements are too generalized and are not supported with references.

10) Table 5: Why aren't the "% burnout" values add up to 100% among different groups? For example: "years of experience" section, "mobbing section", "traumatic event" section, and lastly, "financial situation" section? Please explain.

11) The weaknesses of the study are well articulated. However, the section needs more highlights and further discussion with appropriate references to be cited, as it is a crucial part of the manuscript.

Other suggestions:

1.    The authors are strongly encouraged to use software like Sigma Plot and GraphPad to plot Figures 1 and 2 and increase their overall quality. Moreover, the y-axis labels are missing in both figures, and the figure captions should be comprehensive and explanatory.

2.    The font sizes and types should be consistent across all the figures and tables.

3.    Table spacing across the rows and columns needs to be homogenous.

4.    Please define abbreviations where they appear first in the main text. Please do not use abbreviations in the Abstract.

Author Response

(The authors gave the same response as above.)

Round 2

Reviewer 2 Report

I have no further comments.